# Probiotic and Functional Properties of *Limosilactobacillus reuteri* INIA P572

**DOI:** 10.3390/nu13061860

**Published:** 2021-05-29

**Authors:** Patricia Diez-Echave, Izaskun Martín-Cabrejas, José Garrido-Mesa, Susana Langa, Teresa Vezza, José M. Landete, Laura Hidalgo-García, Francesca Algieri, Melinda J. Mayer, Arjan Narbad, Ana García-Lafuente, Margarita Medina, Alba Rodríguez-Nogales, María Elena Rodríguez-Cabezas, Julio Gálvez, Juan L. Arqués

**Affiliations:** 1Centro de Investigaciones Biomédicas en Red–Enfermedades Hepáticas y Digestivas (CIBER-EHD), Department of Pharmacology, Center for Biomedical Research (CIBM), University of Granada, Avenida del Conocimiento s/n, 18100 Granada, Spain; pdiezechave@gmail.com (P.D.-E.); teresavezza@hotmail.it (T.V.); lhidgar@gmail.com (L.H.-G.); fra.algieri@hotmail.it (F.A.); albarn@ugr.es (A.R.-N.); merodri@ugr.es (M.E.R.-C.); jgalvez@ugr.es (J.G.); 2Instituto de Investigación Biosanitaria de Granada (ibs.GRANADA), 18012 Granada, Spain; 3Departamento Tecnología de Alimentos, INIA-CSIC, Carretera de La Coruña Km 7, 28040 Madrid, Spain; mc.izaskun@gmail.com (I.M.-C.); langa.susana@inia.es (S.L.); landete.josem@inia.es (J.M.L.); mmedina@inia.es (M.M.); arques@inia.es (J.L.A.); 4Gut Microbes and Health Institute Strategic Programme, Quadram Institute Bioscience, Norwich NR4-7UZ, UK; arjan.narbad@quadram.ac.uk (A.N.); melinda.mayer@quadram.ac.uk (M.J.M.); 5Centro para la Calidad de los Alimentos, INIA-CISC, c/José Tudela s/n, 42004 Soria, Spain; ana.glafuente@hotmail.com

**Keywords:** *Limosilactobacillus reuteri*, probiotic, colonic model, DSS-induced colitic model, immunomodulation, reuterin, protective effect

## Abstract

*Limosilactobacillus reuteri* INIA P572 is a strain able to produce the antimicrobial compound reuterin in dairy products, exhibiting a protective effect against some food-borne pathogens. In this study, we investigated some probiotic properties of this strain such as resistance to gastrointestinal passage or to colonic conditions, reuterin production in a colonic environment, and immunomodulatory activity, using different in vitro and in vivo models. The results showed a high resistance of this strain to gastrointestinal conditions, as well as capacity to grow and produce reuterin in a human colonic model. Although the in vitro assays using the RAW 264.7 macrophage cell line did not demonstrate direct immunomodulatory properties, the in vivo assays using a Dextran Sulphate Sodium (DSS)-induced colitic mice model showed clear immunomodulatory and protective effects of this strain.

## 1. Introduction

*Limosilactobacillus reuteri* is a heterofermentative lactobacilli recognized as a normal inhabitant of the human and animal gut [1,2,3,4]. It is frequently found, naturally or added, in a variety of fermented foods, food supplements or infant formulas [1,5,6,7,8]. Some of the probiotic properties attributed to this species include the shortening of infant diarrheal events [9], the decrease in total and LDL-cholesterol levels in hypercholesterolemic subjects [10], the protection against *Helicobacter pylori* infection [11] or the reduction of intestinal inflammation in different experimental models of colitis in rodents [12,13,14,15,16,17]. Moreover, some of these probiotic properties are related to the capacity of certain *L. reuteri* strains to produce the antimicrobial compound reuterin during the anaerobic bioconversion of glycerol [8,18,19] or other metabolites with anti-inflammatory properties [17,20,21]. Recently, it was also demonstrated that reuterin from *L. reuteri* is a functional gut metabolite able to modulate host iron absorption by suppressing intestinal hypoxia-inducible factor 2α activity [22]. Currently, strains of *L. reuteri* are being used as a food supplement to improve gastrointestinal health, and this species has been granted the qualified presumption of safety by the European Food Safety Authority [23]. Daily ingestion of 10^9^–10^10^ alive *L. reuteri* bacteria is considered well tolerated and safe, even in immunodeficient individuals [24,25].

The reuterin-producing *L. reuteri* INIA P572 strain was selected from the INIA culture collection due to its capacity to produce high reuterin yields in milk [26,27,28]. Further experiments showed that this strain was not only able to grow in milk and survive the manufacture and storage conditions during yogurt and cheese elaboration, but also able to produce reuterin in situ in these dairy foods at potentially biologically-active concentrations [27,29]. The bioprotective effect of this strain was further confirmed in cheeses artificially contaminated with the food-borne pathogens *Listeria monocytogenes* and *Escherichia coli* O157:H7 [30] or with *Clostridium tyrobutyricum* strains, responsible for the late blowing defect [31].

Several probiotics, including some strains of *L. reuteri*, have been reported to exert immunoregulatory properties [20,32,33]. These properties are strain-dependent and are attained through diverse mechanisms, from modulating the balance between pro/anti-inflammatory cytokine production upon direct stimulation with the strain [34,35,36] to modulation of intestinal barrier function by decreasing mucosal permeability [37]. In this sense, the viability of a probiotic strain once it reaches the colon is crucial in order to efficiently exert its probiotic function.

Thus, the aim of this study was to look further into the probiotic properties of *L. reuteri* INIA P572 strain, similar to its resistance to gastrointestinal tract conditions, as well as its capacity to grow and produce reuterin in these complex environments. Moreover, the immunomodulatory and protective properties of this strain were tested in vitro using the RAW 264.7 macrophage cell line, and in vivo using a Dextran Sulphate Sodium (DSS)-induced colitis mouse model.

## 2. Materials and Methods

### 2.1. Bacterial Strains and Growth Conditions

The reuterin-producing *L. reuteri* INIA P572 was selected from the INIA culture collection due to its high reuterin yields [26]. The strain was propagated in Man Rogosa Sharpe (MRS) broth (Biolife, Milano, Italy) at 37 °C and anaerobic conditions (Anaerogen TM, Oxoid, Basingstoke, UK). Transformed *L. reuteri* INIA P572 with pNZ:TuR.aFP (*L. reuteri* INIAP572:aFP), expressing the anaerobic fluorescence protein gene evoglow-Pp1 [38], was propagated in MRS with 8 µg/mL chloramphenicol (Cm) (Sigma-Aldrich, Madrid, Spain). For microbiological analysis, serial dilutions of *L. reuteri* INIA P572 were plated in Rogosa agar, while serial dilutions of *L. reuteri* INIA P572:aFP were plated in Rogosa agar with 8 µg/mL Cm.

### 2.2. Resistance to Stimulated Gastrointestinal Conditions In Vitro

Resistance to gastrointestinal conditions was studied in vitro in both parental *L. reuteri* INIA P572 and *L. reuteri* INIAP572:aFP based on the method described by Haller et al. [39] with some modifications. Both strains were grown overnight as described previously and centrifuged at 7000× *g* for 5 min. Pellets were resuspended in UHT semi skimmed milk and initial counts were performed. The milk solution was diluted 1:10 in PBS and adjusted to pH 3. Bacterial counts were carried out after 1 h of incubation. Then, Oxgall (Oxoid) was added to the samples at 0.15% (pH 8.0) and these were incubated for 1 h at 37 °C and anaerobic conditions. Bacterial counts were also performed. All assays were done in duplicates.

### 2.3. Viability of L. reuteri INIA P572 and Reuterin Production in an In Vitro Colonic Model. Modulation of Fecal Bacterial Population

Reuterin production was assessed in an in vitro colonic model. Growth media was prepared as described by Vulevic et al. [40] and distributed in different batch fermenters (135 mL). Fresh faecal samples from a healthy human volunteer, with no history of antibiotics treatment in the previous 6 months, were first diluted in 0.1 M PBS pH 7.4, homogenized, filtered, and distributed in each of the batch fermenters (15 mL). The first batch fermenter was inoculated with *L. reuteri* INIA P572, the second was inoculated with 100 mM glycerol, while the remaining one was inoculated with *L. reuteri* INIA P572:aFP and 100 mM glycerol. The batch fermenters were stirred and maintained under anaerobic conditions. The temperature was set at 37 °C and pH was maintained between 6.6 and 7.0, using a pH controller (Electrolab, UK). Three separate fermentation experiments were carried out and two independent samples were taken from each batch fermenter at 0, 6, and 24 h.

For the microbiological analysis, samples from the different batches were serially diluted in peptone water and counts were performed for the main microbial groups normally present in faeces: Aerobic, anaerobic, *Bacteroidaceae*, *Clostridiaceae*, *Enterobacteriaceae*, *Lactobacillaceae*, as described by Vulevic et al. [40]. *L. reuteri* INIA P572 counts were carried out as explained previously. For reuterin detection, samples were centrifuged at 12,000× *g* for 20 min at 4 °C and supernatants were sterilized by filtering (0.22 µm, Millipore Corporation, Bedford, MA, USA) and frozen at -20 °C until analysis. Reuterin production was determined in supernatants by the colorimetric method of Circle et al. [41] as explained by Langa et al. [27], against a standard curve with known concentrations of purified reuterin in fermenter growth media. The presence of reuterin was further confirmed by ^1^H NMR analysis. Briefly, 100 μL samples were mixed with a 900 μL NMR buffer (0.26 g NaH_2_PO_4_ and 1.41 g K_2_HPO_4_ made up in 100 mL D2O, containing 0.1% NaN_3_ (100 mg), and 1 mM sodium 3-(Trimethylsilyl)-propionate-d4, (TSP) (17 mg) as a chemical shift reference. The supernatant was analyzed using 1 H NMR spectroscopy, recorded at 600 MHz on a Bruker Avance spectrometer (Bruker BioSpin GmbH, Rheinstetten, Germany) with a cryoprobe, a 60-slot autosampler, and the software Topspin 2.0. Each 1 H NMR spectrum was acquired with 256 scans, an acquisition time of 2.67 s, and a spectral width of 12,300 Hz. To suppress the residual water signal, the “noesypr1d” pre-saturation sequence was used, with a low-power selective irradiation at the water frequency during the recycle delay and a mixing time of 10 ms. In addition, the 0.3 Hz line broadening transformed spectra were manually phased, baseline corrected, and referenced by setting the TSP methyl signal to 0 ppm. The Chenomx^®^ NMR Suite 7.0 software was used for metabolite quantification.

### 2.4. Immunomodulatory Activity In Vitro

The immunomodulatory activity of *L. reuteri* INIA P572 was studied in vitro using the murine macrophage cell line RAW 264.7 (European Collection of Authenticated Cell Cultures; ECACC, Salisbury, UK). RAW 264.7 cells were cultured in Dulbecco’s Modified Eagle’s Medium (DMEM) supplemented with heat inactivated fetal bovine serum (FBS), 100 IU/mL penicillin, and 100 µg/mL streptomycin (all from Sigma-Aldrich). Cells were routinely cultured at 37 °C in a HF160W incubator (Heal Force, Burwood, Australia) with a humidified 5% CO_2_ atmosphere. *L. reuteri* INIA P572 bacterial suspension was prepared in DMEM with gentamicin (Lonza, Barcelona, Spain) at 10^5^ cfu/mL. Before cell stimulation, RAW 264.7 cells were scraped, cell suspension was adjusted to 10^6^ cells/mL and 200 µL of the suspension was incubated in each well in a 96-well plate, and were incubated for 18 h before the experiments.

Cell viability of RAW 264.7 cells after *L. reuteri* INIA P572 incubation was assessed by the colorimetric MTT (3-(4,5-dimethylthiazol-2-yl)-2,5-diphenyltetrazolium bromide) assay (Sigma-Aldrich) as previously described [42]. Cell viability (%) was calculated by comparing sample absorbance values with untreated control cultures.

#### 2.4.1. Nitric Oxide (NO) Production

NO production was studied in supernatants from stimulated RAW 264.7 cells with *L. reuteri* INIA P572 or with lipopolysaccharides (LPS) from *Escherichia coli* (Sigma-Aldrich) (0.01, 0.1 or 1 µg/mL). In the assays with the bacterial strain, some of the wells were also stimulated with 1 μg/mL LPS after 1 h. After 24 h of incubation, plates were centrifuged and supernatants were frozen until further analysis.

NO produced by RAW 264.7 macrophage cells was quantitatively analyzed in supernatants by the Griess reaction [43,44,45]. Nitrite concentration in supernatants was calculated against a sodium nitrite standard curve (2.5, 5, 10, 20, and 50 mM) in fresh DMEM media. For *L. reuteri* plus LPS assays, data were expressed as percentage, considering the average NO production in cells stimulated with LPS as the 100%.

#### 2.4.2. Cytokine Production

Cytokine production was studied in supernatants from RAW 264.7 cells stimulated or not with *L. reuteri* INIA P572. After 1 h, 1 μg/mL LPS was added to some of the wells. After 24 h of incubation, plates were centrifuged and supernatants were frozen until further analysis. Tumor necrosis factor-α (TNF-α), interleukin-1β (IL-1β), and interleukin-6 (IL-6) levels were determined using the mouse Duo-set ELISA kits according to the R&D systems protocols (Minneapolis, MN, USA).

### 2.5. Animal Studies

All animal studies were in accordance with the Guide of the Care and Use of Laboratory Animals and the protocols approved by the Ethic Committee of Laboratory Animal of the University of Granada (Spain) (Ref. no. CEEA 17/09/2019/156).

### 2.6. Dextran Sodium Sulfate (DSS) Model of Mouse Colitis

The in vivo dextran sodium sulfate (DSS) model of mouse colitis was used as described by Garrido-Mesa et al. 2015. Male C57BL/6J mice (7–9 weeks; approximately 20 g) were obtained from Janvier Labs (St Berthevin Cedex, France) and maintained in an air-conditioned atmosphere with a 12 h light/dark cycle, and free access to tap water and food. Mice were randomly divided into 3 experimental groups of 10 animals, and two of the groups were rendered colitis by adding (3% *w*/*v*) DSS (36–50 kDa, MP Biomedicals, Ontario, USA) in the drinking water for 5 days. Mice were daily treated by oral gavage. Non-colitic and DSS-colitic groups were included as a reference, and received the vehicle (semi-skimmed milk; 200 µL/mouse/day). The treated group was administered the probiotic *L. reuteri* INIA P572 (5 × 10^8^ log cfu/mL in 200 µL vehicle/mouse/day). All treatments began 14 days before colitis induction and continued until the sacrifice of mice, 24 days later.

#### 2.6.1. Intestinal Inflammatory Process Evaluation

Animal body weight, gross rectal bleeding, and stool consistency were assessed daily over a 6-day period by an observer unaware of the treatment. These parameters were scored according to the previously proposed criteria [46] and this information was used to calculate the average daily disease activity index (DAI). At the end of the experiment, animals were sacrificed, the colon was removed and its length and weight were measured under a constant load (2 g). Whole gut specimens (0.5 cm length) were taken from the distal region, representative of DSS-damage, and fixed in 4% buffered formaldehyde for the histological studies. The remaining colonic tissue was subsequently sectioned in longitudinal fragments for RNA and lamina propria immune cell isolations.

#### 2.6.2. Histological Analysis

Five µm sections were obtained from paraffin-embedded colonic specimens and stained with hematoxylin, eosin, and alcian blue. A pathologist observer, blinded to the experimental groups, scored the histological damage according to the previously described criteria [47]. Oedema, ulceration, infiltration, and the condition of the crypts were evaluated, scored from 0 (healthy) to 3 or 4 (severe damage). The total score for each specimen was calculated as the sum of those values.

#### 2.6.3. Analysis of Gene Expression in Colonic Samples by RT-qPCR

The colonic RNA content was extracted using the RNeasy^®^ MiniKit (Qiagen, Hilden, Germany), following the manufacturer’s instructions. All RNA samples were quantified using a Thermo Scientific NanoDrop^TM^ 2000 Spectrophotometer (Thermo Scientific, Wilmington, DE, USA). In addition, 2 µg RNA samples were reverse transcribed using oligo (dT) primers (Promega, Southampton, UK). Real-time quantitative PCR amplification and detection was performed in a 7500 RT-PCR System (PE Applied Biosystems, CA, USA) as previously described [48] using specific primers (Table 1). Normalization of mRNA expression was performed using the housekeeping gene glyceraldehyde-3-phosphate dehydrogenase (GAPDH) and the relative expression level was calculated using the ΔΔCt method.

#### 2.6.4. Flow Cytometry

Colonic lamina propria (cLP) infiltrating cells were isolated as described previously [49] using a digestion media composed of HBSS without Mg2+ or Ca2+, 10% of FBS and 0.5 mg/mL collagenase V (Sigma-Aldrich, St. Louis, MO, USA), 0.65 mg/mL collagenase D, 30 μg/mL DNase I and 1 mg/mL dispase II (Roche Diagnostics Gmb-H, Mannheim, Germany). The FcR blocking reagent (Miltenyi, Pozuelo de Alarcón, Madrid, Spain), surface-staining antibodies and viability stain (Invitrogen, Carlsbad, CA, USA) were added to the cell suspension and incubated for 20 min at 4 °C. For intracellular cytokine expression, cells were pre-stimulated with PMA (50 ng/mL) and ionomycin (1 μg/mL) (Sigma-Aldrich, St. Louis, MO, USA) in the presence of GolgiPlug™ (eBioscience, Thermo Fisher, Carlsbad, CA, USA) for 4.5 h at 37 °C. After surface staining, cells were fixed with a Fixation/Permeabilization buffer (FoxP3 Staining Kit, eBioscience), and intracellular staining was performed following the manufacturer’s instructions. The antibodies α-mouse were from Miltenyi unless otherwise stated: CD45 (30F11), CD11b (M1/70.15.11.5), Ly6G (REA526), SiglecF (REA798), F4/80 (REA126), CD11c (N418), B220 (RA3–6B2, BD Bioscience, San Jose, CA, USA), CD3 (17A2), CD8 (53–6.7), CD4 (RM4–5, BD Bioscience), IL-4 (BVD4-1D11), IFNγ (XMG1.2, BD Pharmigen), IL-17A (eBio17B7, eBioscience), and FoxP3 (FJK-16s, eBioscience). Samples were acquired in a FACSVerse™ or FACSCanto II™ cytometer (Becton Dickinson, Franklin Lake, NJ, USA), and data were analysed with the FlowJo software (Tree Star, Woodburn, OR, USA). Each population percentage over live cells was multiplied by the total count to obtain total cell numbers.

### 2.7. Statistical Analysis

Data were subjected to ANOVA with the SPSS program 22.0 for Windows (IBM corp., Armonk, NY, USA) using a general linear model. The comparison of means was assessed by the Student’s t-test (comparison of pairs) or by Tukey´s multiple range test at *p* < 0.05.

## 3. Results

### 3.1. L. reuteri INIA P572 Resistance to Gastrointestinal Conditions In Vitro

*L. reuteri* INIA P572 and *L. reuteri* INIA P572:aFP strains showed a high resistance to pH 3-simulated gastric conditions, with bacterial counts recovered only one log unit below than the initial level. Similarly, bacterial exposure to small intestine-like conditions, pH = 8 in the presence of bile salts, had an impact lower than 0.3 log units from the initial concentration. No significant differences were observed between the two strains (data not shown).

### 3.2. Viability of L. reuteri INIA P572 and Faecal Microbiota in an In Vitro Colonic Model. Reuterin Production

Faecal microbiota counts showed the same trend upon time in the three batch fermenters (Table 2), increasing their level to 8–9 log units approximately after 24 h of incubation in all the microbial groups, with the exception of lactobacilli that reached levels of 7 log units. No significant differences in counts were observed between batch fermenters in any of the microbial groups studied at the same time point. No influence of *L. reuteri* INIA P572 and/or 100 mM glycerol in the faecal microbiota levels was observed.

The viability of *L. reuteri* INIA P572:aFP in an in vitro colonic model was evaluated (Table 2). Counts of INIA P572 were significantly (*p* < 0.05) influenced by the time of incubation in the batch fermenters, but not by the addition of glycerol. In both batch fermenters, inoculated only with *L. reuteri* INIA P572:aFP or plus glycerol, an increase in the probiotic strain levels lower than 1 log unit was observed after 6 h.

Reuterin production in the three batch fermenters was initially determined by the Circle colorimetric method in samples taken at 0 and 24 h. At 0 h, no reuterin was detected by this method in any of the batch fermenters. However, after 24 h, reuterin was detected only in samples from the batch fermenters inoculated with *L. reuteri* INIA P572 and 100 mM glycerol. In this method, the fermentation media was interfering with absorbance at 490 nm, and only reuterin concentrations higher than 2 mM could be detected in the samples. The presence of reuterin in positive samples was confirmed by ^1^H-NMR spectroscopy, in which ^1^H-NMR spectra for reuterin were recorded in 24 h samples from batch fermenters inoculated with *L. reuteri* INIA P572:aFP and 100 mM glycerol (data not shown). In contrast, no reuterin was detected by this technique in samples taken at 0 h from any of the batch fermenters and at 24 h from batch fermenters with only *L. reuteri* INIA P572:aFP or 100 mM glycerol.

### 3.3. Immunomodulatory Activity of L. reuteri INIA P572 Using In Vitro Models

The immunomodulatory activity of *L. reuteri* INIA P572 was studied in RAW 264.7 cells by measuring NO production upon 24 h of stimulation with the probiotic, and compared with different concentrations of LPS (Figure 1A). NO production ranged between 7.81 and 24.86 µM for LPS concentrations from 0.01 to 1 µg/mL. When RAW 264.7 cells were incubated with *L. reuteri* INIA P572, NO production was 2.64 µM. Cell viability upon stimulation with *L. reuteri* INIA P572 was also studied. MTT assays showed that RAW 264.7 cell viability was not affected by the presence of *L. reuteri* INIA P572 (data not shown).

In addition, the effect of *L. reuteri* INIA P572 was also studied in RAW 264.7 cells stimulated with 1 µg/mL LPS. The incubation with *L. reuteri* INIA P572 before LPS stimulation did not significantly modify NO production in comparison with those cells treated with LPS only (Figure 1B).

The impact of the probiotic *L. reuteri* INIA P572 on cytokine production was analyzed in RAW 264.7 cells (Figure 2). The basal production of TNF-α, IL-1β, and IL-6 by macrophages was not significantly modified in the presence of *L. reuteri* INIA P572. However, the incubation of these cells with LPS resulted in significant increased levels of these cytokines in comparison with basal conditions (*p* < 0.05), which were even higher when the cells were pretreated with *L. reuteri* INIA P572, although statistical differences were only obtained when IL-1β production was considered.

### 3.4. Intestinal Anti-Inflammatory Effect of L. reuteri INIA P572 in Experimental Colitis

The addition of DSS 3% (*w*/*v*) in the drinking water induced an intestinal inflammatory process, characterized by body-weight loss, diarrhoea, and rectal bleeding. Consequently, DAI values increased upon DSS administration, but the probiotic treatment ameliorated DSS-colitis development compared to the untreated control mice (Figure 3). The histological analysis of colonic cross-sections corroborated the beneficial effect obtained in colitic mice with the administration of *L. reuteri* INIA P572 (Figure 4). The microscopic impact of DSS-colitis in untreated control mice was characterized by epithelial ulceration, crypt hyperplasia, depletion of mucin-containing goblet cells, and inflammatory cell infiltration into the lamina propria, with a median (range) microscopic score of 32 (28–39). Of note, DSS-colitic mice treated with the probiotic showed a significant restoration of the mucosal epithelium, with restoration of mucin-producing goblet cells and lower inflammatory-cell infiltration in the lamina propria. Therefore, the administration of *L. reuteri* INIA P572 resulted in a significant reduction of the microscopic values compared to the DSS-colitic group, with a median (range) microscopic score of 14 (8–23).

The intestinal anti-inflammatory probiotic effect was also corroborated when the expression of different intestinal inflammatory markers was analyzed. Increased expression of pro-inflammatory cytokines, Il-1β, Il-6, and TNF-α, monocyte chemotactic protein-1 (Mcp-1) and macrophage inflammatory protein 2 (Mip-2), characterized DSS-induced colonic inflammation compared to non-colitic mice (Figure 5). *L. reuteri* INIA P572 pretreatment counteracted the upregulation of these pro-inflammatory mediators. Furthermore, Il-6 and Mip-2 expression were restored to the baseline levels of non-colitic mice. Inducible enzyme nitric oxide synthase (Inos) and the intracellular adhesion molecule-1 (Icam-1) were also significantly upregulated in DSS-colitic mice when compared to non-colitic mice, which were significantly downregulated in probiotic-treated colitic mice (Figure 5).

To further explore the immunomodulatory effect of *L. reuteri* INIA P572, colonic lamina propria immune infiltration was evaluated by flow cytometry. Total immune (CD45^+^) cell infiltration was reduced by the probiotic treatment when compared to control DSS mice, which was associated with a decrease in both the myeloid (CD11b^+^) and lymphoid cells (Figure 6 and Figure 7). Among the myeloid compartment (Figure 6), the treatment with *L. reuteri* INIA P572 significantly reduced the number of macrophages (F4/80^+^), dendritic cells (CD11c^+^MHCII^+^), neutrophils (Ly6G^+^), and eosinophils (SiglecF^+^). When considering the lymphoid cells (Figure 7), B cell infiltration was also ameliorated by the *L. reuteri* INIA P572 probiotic treatment. Of note, the T cell response to the colitis process was totally blunted in the treated group, an effect evidenced across different T cell subpopulations, such as cytotoxic CD8^+^ T cells, Th1 (CD4^+^IFNg^+^), Th2 (CD4^+^IL4^+^), Th17 (CD4^+^IL17^+^), and Tregs (CD4^+^FoxP3^+^).

## 4. Discussion

Probiotic strains should remain metabolically active after the gastrointestinal passage in order to exert its probiotic action in the gut. This is crucial to impact the balance of the intestinal microbiota [51] and, in many cases, also to confer stimulation of the intestinal immunity [52]. In this study, *L. reuteri* INIA P572 has shown high resistance to gastrointestinal conditions in experiments performed in vitro, reinforcing its potential as a probiotic microorganism. Similar assays have been performed previously as a screening tool for new probiotic strains [26,53,54], as the resistance to gastrointestinal passage is strain dependent, similar to other probiotic properties [55,56]. As *L. reuteri* INIA P572 would reach the gut at high numbers upon consumption, we studied if the conditions in the colon would allow its survival and the production of reuterin. In the in vitro colonic model, *L. reuteri* INIA P572 levels increased significantly after 6 h and remained stable for 24 h in the presence or absence of 100 mM glycerol and this did not have any influence in its counts at any time. Similar to previous reports [57], the addition of *L. reuteri* INIA P572 and/or 100 mM glycerol in this model did not have a negative influence in the number of different groups of intestinal microbiota, suggesting that there is no negative effect on the indigenous microbiota. In situ reuterin production was detected by the colorimetric method only in the batch fermenters inoculated with *L. reuteri* INIA P572 and glycerol after 24 h of incubation and this was further confirmed by ^1^H-NMR. In situ reuterin production during the manufacture and storage of different dairy foods at similar conditions has already been described [27,29]. Similar studies have looked into reuterin production by *L. reuteri* ATCC PTA 6475 in an intestinal model adding 40 and 140 mM glycerol, however its presence was detected indirectly as 1,3-propanodiol, postulating the production of reuterin [58]. Different bacteria genera (*Klebsiella*, *Citrobacter*, *Clostridium,* and *Limosilactobacillus*) are able to produce reuterin as an intermediate metabolite during glycerol anaerobic bioconversion. Most of these species immediately reduce this intermediate product using NAD-dependent oxidoreductases, obtaining 1,3 PDO as a final product [59]. However, some lactobacilli strains are able to accumulate and excrete reuterin extracellularly [26,60,61,62,63], with *L. reuteri* being among the most efficient producers of reuterin, although not all its strains exhibit this ability [28]. Morita et al. [64] have already shown reuterin production ex vivo in mice intestine by a *L. reuteri* strain in the presence of 7–10 mM glycerol. Similar studies with a model of colonic epithelium indicated a potential role for reuterin in inhibiting *Salmonella*-induced intestinal infections, showing also that the short-term exposure of *L. reuteri* ATTC PTA6475 and 100 mM glycerol did not have a negative effect on the colonic epithelial cells, displaying similar numbers upon reuterin production [65]. Several studies have looked into the main bacterial groups present in the faecal microbiota of these fermentation models and the influence of different prebiotic compounds in its composition [66,67,68]. In this work, the presence of *L. reuteri* INIA P572 and/or 100 mM glycerol has no influence in the composition of these groups at any time, suggesting that the addition of this strain with or without glycerol will not dramatically affect the host microbiota.

In the present study, immunomodulatory properties of *L. reuteri* INIA P572 were assessed both in vitro and in vivo. The RAW 264.7 macrophage cell line has been frequently used in studies evaluating the immunomodulatory activity of probiotic strains in vitro [47,69,70,71]. The initiation and the regulation of innate inflammatory responses to intestinal bacteria, particularly to LPS, one of the most important pathogen-associated molecular patterns, is tightly controlled by macrophages [72], which trigger the release of pro-inflammatory cytokines TNF-α, IL-1β, and IL-6 by macrophages [72]. Additionally, activated macrophages induce the expression of NO synthase (NOS), which results in NO production, one of the cytotoxic agents involved in their inflammatory response, but also in autoimmune reactions [73]. In our experiments, RAW 264.7 cells showed a very low NO production upon stimulation with *L. reuteri* INIA P572 in comparison with LPS (0.01–1 µg/mL), as previously reported with other lactobacilli [69]. However, pre-incubation for 1 h did not reduce the production of NO after stimulation with LPS (1 µg/mL). Previous in vitro studies have investigated the RAW 264.7 macrophage responses to lactobacilli strains [70,74], concluding that the inflammatory profile elicited by these bacteria depends on LPS concentration [75]. Of note, TNF-α, IL-1β, and IL-6 are produced by macrophages in different inflammatory conditions, including inflammatory bowel diseases (IBD) [76]. In the present study, the exposure of macrophages to bacterial LPS promoted a higher secretion of the cytokines TNF-α, IL-1β, and IL-6 when compared with the incubation of *L. reuteri* INIA P572. Moreover, this strain did not induce significant differences in the production of TNF-α, IL-6, and IL-1β when the cells were treated with 1 µg/mL LPS, as reported with other lactobacilli [77]. In fact, previous studies using non-stimulated RAW 264.7 cells or similar models have reported the ability of different lactobacilli and bifidobacteria strains to stimulate the production of pro-inflammatory cytokines [78,79,80,81], although the mechanisms implicated remain unknown. In contrast, experiments performed by Pena and Versalovic [82] revealed the reduction of cytokine production by *L. rhamnosus GG* in RAW 264.7 cells when stimulated with LPS, but using a concentration almost 10-fold lower (2 ng/mL) than that used in the present study. These results suggest that the ability of a probiotic to modulate the production of pro-inflammatory cytokines may decrease as the LPS concentration is increased.

To evaluate its resilience to intestinal passage and immunomodulatory potential, *L. reuteri* INIA P572 was assayed in the well-established DSS experimental model of colitis, where the intestinal inflammation resembles that found in human IBD. The results revealed that *L. reuteri* INIA P572 exerted an intestinal anti-inflammatory effect clearly related to its immunomodulatory properties, as supported by gene expression and cytometry analysis. Previous studies with other strains of *L. reuteri* have also reported beneficial effects in experimental colitis, both in mice and rats [83]. The beneficial effects observed with the preventative administration of *L. reuteri* INIA P572 were initially evidenced by a significant reduction of DAI, indicative of the protective effect against DSS-induced colonic pathology, both at the macroscopic and microscopic level. In fact, the DAI evolution has been considered as a suitable marker to evaluate the efficacy of different treatments in reducing colonic inflammation in experimental models of colitis in rodents [46,84,85,86]. The histological assessment revealed that probiotic administration reduced mucosal damage as well as the inflammatory cell infiltration in the colonic mucosa. The latter was corroborated by the flow cytometry assays, since the administration of *L. reuteri* INIA P572 to colitic mice ameliorated the infiltration of all the immune populations evaluated. This effect correlates with the decreased expression of proinflammatory mediators, which might anticipate a reduction in immune recruitment and subsequent tissue damage. The opposite hypothesis might also explain the beneficial probiotic effect, particularly considering the lack of anti-inflammatory actions observed in macrophages in vitro: A protective impact on the mucosal/stromal component that leads to reduced damage and immune recruitment. However, considering the complexity of the intestinal immune response and the pleiotropic effect of probiotics, a deeper profiling would be required to ascertain the underlying mechanism.

The differences in the results obtained using in vitro and in vivo models to test the same strain are not unusual, as other authors have also observed discordances in the data obtained from in vitro and in vivo studies for the same probiotic strain [87,88]. In vitro studies do not mimic important factors that are present in animal models, such as the resident microbiota and the barrier function of the intestinal mucosa, distanced from physio/pathological settings [71]. Studies performed in human colonic microbiota models showed that colonic LPS concentrations were positively and significantly correlated with TNF-α and IL-1β levels. This suggests that specific probiotic strains could decrease colonic LPS and, therefore, pro-inflammatory cytokine levels [71]. In addition, previous ex vivo and in vivo studies have shown that lactobacilli strains have been successfully used to modulate inflammatory diseases and enhance the barrier function [89,90].

## 5. Conclusions

In conclusion, the probiotic strain *L. reuteri* INIA P572 is able to resist gastrointestinal and colonic conditions, produce reuterin in the colonic environment, and exert an immunomodulatory/protective role in DSS-induced colitis mice. These findings complement previous studies and reinforce the application of this strain as a commercial probiotic. This strain may also be suitable for developing novel multi-strain probiotic food products with complementary positive effects in both food safety and maintaining gut homeostasis.

## Figures and Tables

**Figure 1 nutrients-13-01860-f001:**
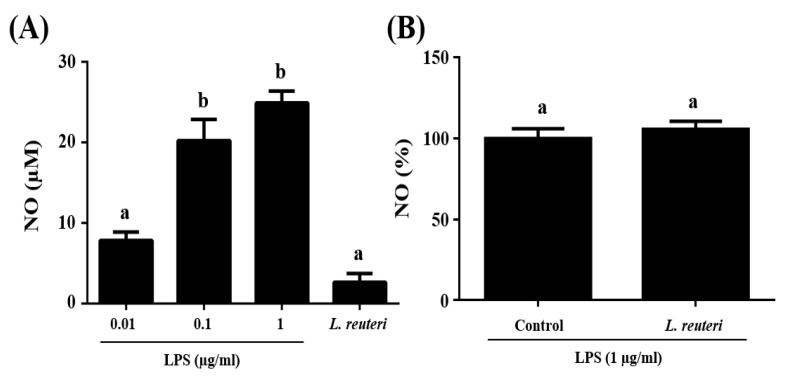
Data represent the mean ± standard deviation of at least three separate experiments. (**A**) NO production (µM) in a RAW 264.7 cell line upon stimulation with lipopolysacharides (LPS) (0.01, 0.1, and 1 µg/mL) or *L. reuteri* INIA P572. (**B**) Nitric Oxide (NO) production (%) of RAW 264.7 cells stimulated only with 1 µg/mL LPS (considered 100%) or previously activated with *L. reuteri* INIA P572. Different samples with different letters (a-c) show significant differences (*p* < 0.05).

**Figure 2 nutrients-13-01860-f002:**
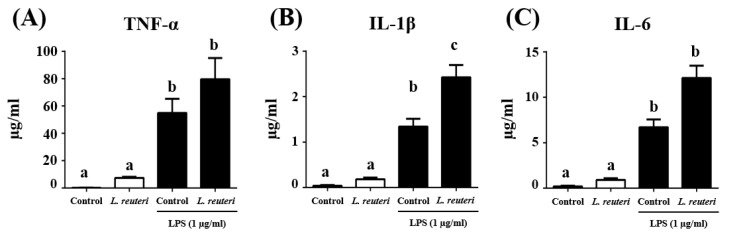
Tumor necrosis factor-α (TNF-α) (**A**), interleukin-1β (IL-1β) (**B**), and interleukin-6 (IL-6 ) (**C**) levels (µg/mL) in RAW 264.7 cells, untreated (white) or treated with 1 µg/mL lipopolysaccharides (LPS) (black), and/or pre-stimulated with *L. reuteri* INIA P572. Data represent the mean ± standard error of the mean. Different samples with different letters (a–b) show significant differences (*p* < 0.05).

**Figure 3 nutrients-13-01860-f003:**
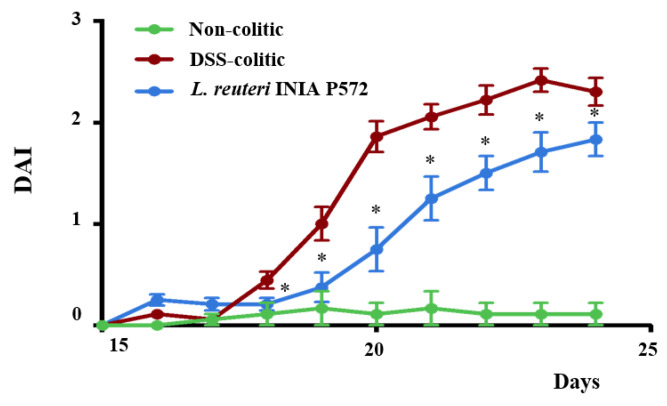
Effects of *L. reuteri* INIA P572 on disease activity index (DAI) values in Dextran Sulphate Sodium (DSS) colitic mice over the experimental period. Data are expressed as mean ± SEM (*n* = 10). Both colitic groups significantly differ from the non-colitic group (*p* < 0.05). * *p* < 0.05.

**Figure 4 nutrients-13-01860-f004:**
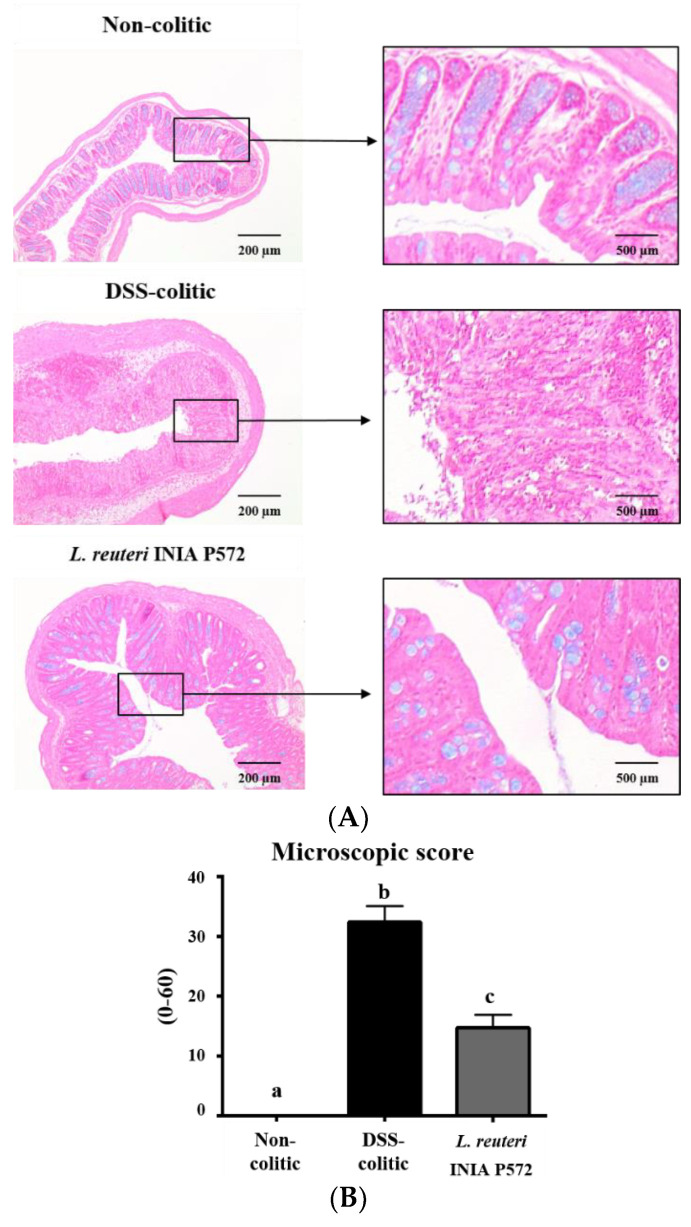
Impact of *L. reuteri* INIA P572 on (**A**) histological sections of colonic tissue stained with hematoxylin, eosin, and alcian blue. (**B**) Microscopic score assigned to each group according to the criteria previously described [50]. Data (*n* = 10 per group) are expressed as mean ± SEM; groups with different letters statistically differ (*p* < 0.05).

**Figure 5 nutrients-13-01860-f005:**
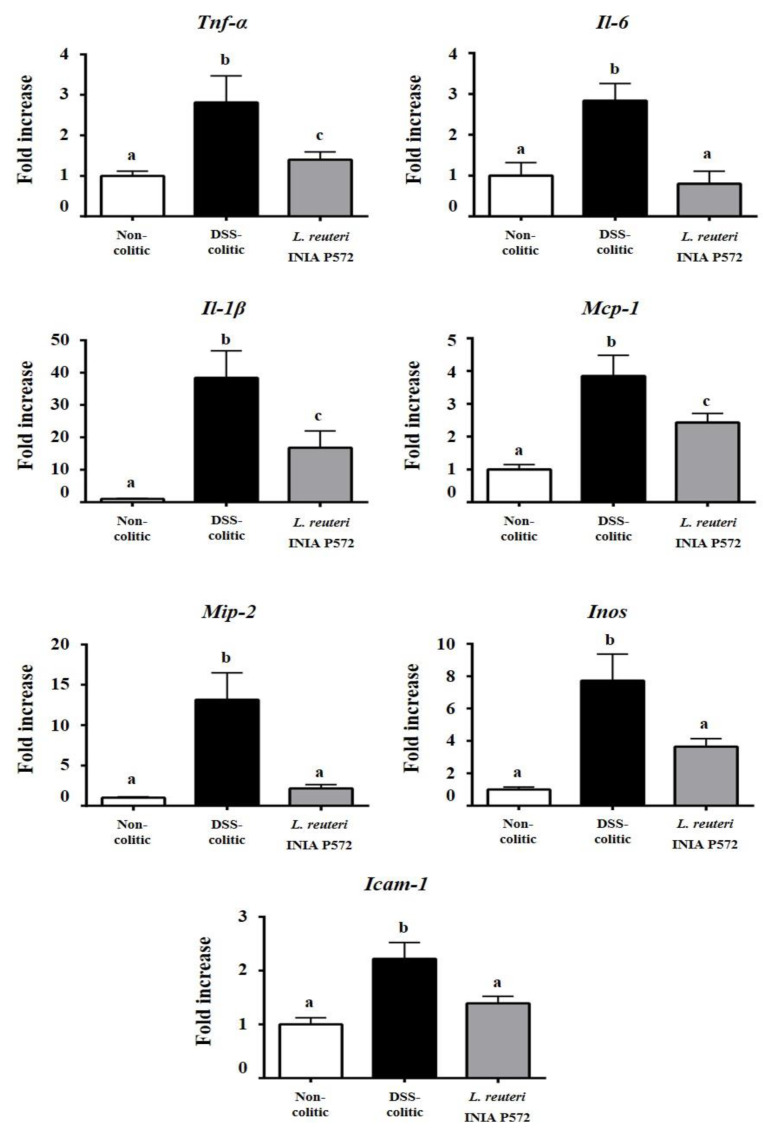
Effects of *L. reuteri* INIA P572 on colonic gene expression of TNF-α, IL-6, IL-1β, monocyte chemotactic protein-1 (Mcp-1), macrophage inflammatory protein 2 (Mip-2), Inducible enzyme nitric oxide synthase (Inos), and intracellular adhesion molecule-1 (Icam-1) in DSS-colitic mice, analyzed by real-time qPCR. Data are expressed as mean ± SEM (*n* = 10). Groups with different letters statistically differ (*p* < 0.05).

**Figure 6 nutrients-13-01860-f006:**
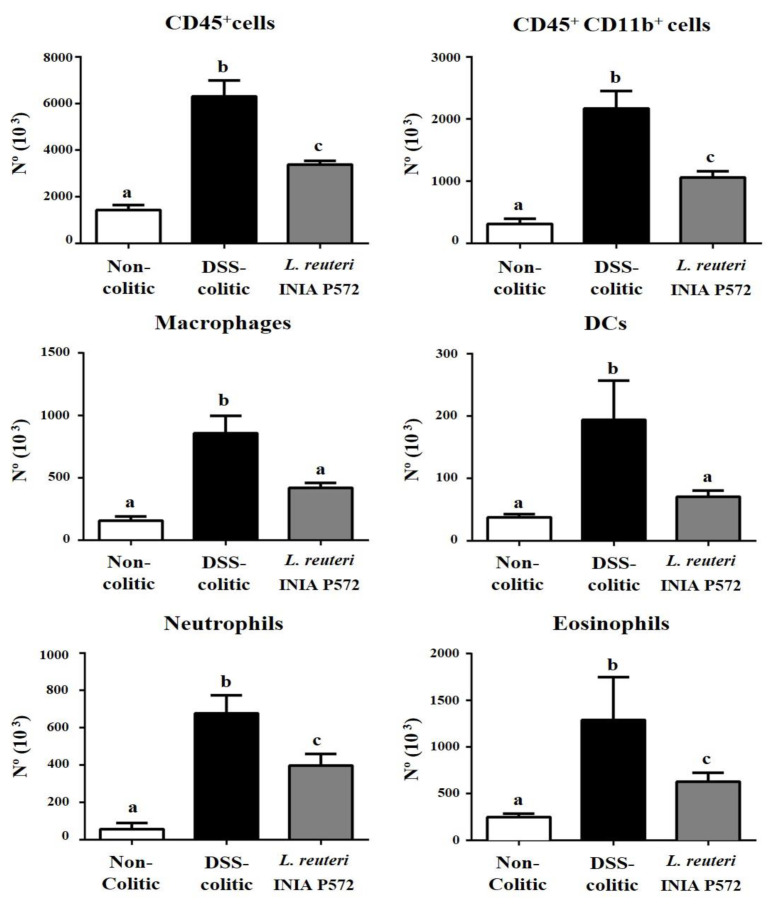
Evaluation of the effect of *L. reuteri* INIA P572 on the immune response in DSS-colitic mice. Analysis of the indicated immune cell population in the colonic lamina propria of the different experimental groups. Absolute cell numbers of immune cells (CD45+), lymphoid cells (CD11b+), macrophages (F4/80+), dendritic cells (CD11c+ MHCII+), neutrophils (LY6G+), and eosinophils (SiglecF+). Data are expressed as mean ± SEM (*n* = 10). ^a,b,c^ Groups with different letters statistically differ (*p* < 0.05).

**Figure 7 nutrients-13-01860-f007:**
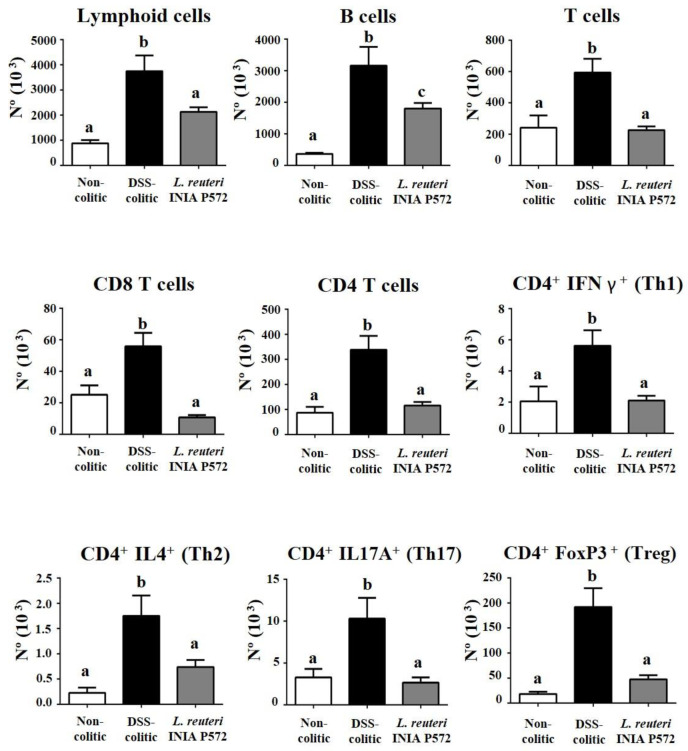
Evaluation of the impact of *L. reuteri* INIA P572 on the immune response in DSS-colitic mice. Analysis of the indicated immune cell population in the colonic lamina propria of the different experimental groups. Absolute cell number (N°) of lymphoid cells, B cells (B220+), T cells, cytotoxic T cells (CD8+), CD4 T cells, Th1 cells (CD4+ IFNγ+), Th2 cells (CD4+ IL4+), Th17 (CD4+ IL17+), and Tregs (CD4+ FoxP3+). Data are expressed as mean ± SEM (*n* = 10). ^a,b,c^ Groups with different letters statistically differ (*p* < 0.05).

**Table 1 nutrients-13-01860-t001:** Primer sequences and annealing temperatures used in real-time PCR assays in colonic tissue.

Gene	Sequence (5′-3′)	Annealing Temperature (°C)
GAPDH	FW:CCATCACCATCTTCCAGGAG RV:CCTGCTTCACCACCTTCTTG	60
TNF- α	FW:CCATCACCATCTTCCAGGAG RV:CTTCACAGAGCAATGACTCC	56
IL-6	FW:TAGTCCTTCCTACCCCAATTTCC RV:TTGGTCCTTAGCCACTCCTTC	60
IL-1β	FW:TGATGAGAATGACCTCTTCT RV:CTTCTTCAAAGATGAAGGAAA	55
MCP-1	FW:CAGCTGGGGACAGAATGGGG RV:GAGCTCTCTGGTACTCTTTTG	62
MIP-2	FW:CAGTGAGCTGCGCTGTCCAATG RV:CAGTTAGCCTTGCCTTTGTTCAG	60
iNOS	FW:GTTGAAGACTGAGACTCTGG RV:GACTAGGCTACTCCGTGGA	56
ICAM-1	FW:GAGGAGGTGAATGTATAAGTTATG RV:GGATGTGGAGGAGCAGAG	60

**Table 2 nutrients-13-01860-t002:** Bacterial numbers (log cfu/mL) in samples from batch fermenters inoculated with *L. reuteri* INIA P572 (INIA P572), 100 mM glycerol (glycerol), and *L. reuteri* INIA P572 with glycerol (INIA P572 glycerol).

	t (h)	INIA P572	Glycerol	INIA P572 Glycerol
Total aerobes	0	7.19 ± 0.12 ^aA^	7.30 ± 0.10 ^aA^	7.38 ± 0.10 ^aA^
	6	9.13 ± 0.23 ^bA^	9.19 ± 0.07 ^bA^	8.83 ± 0.43 ^bA^
	24	9.32 ± 0.05 ^bA^	9.08 ± 0.26 ^bA^	9.44 ± 0.16 ^bA^
Anaerobes	0	7.58 ± 0.18 ^aA^	7.34 ± 0.10 ^aA^	7.46 ± 0.08 ^aA^
	6	9.16 ± 0.13 ^bA^	9.12 ± 0.14 ^bA^	9.15 ± 0.23 ^bA^
	24	9.31 ± 0.09 ^bA^	9.13 ± 0.31 ^bA^	9.45 ± 0.10 ^bA^
Bacteroides	0	6.85 ± 0.24 ^aA^	6.57 ± 0.07 ^aA^	7.13 ± 0.40 ^aA^
	6	7.12 ± 0.04 ^aA^	6.63 ± 0.22 ^aA^	7.41 ± 0.21 ^aA^
	24	8.45 ± 0.10 ^bA^	8.26 ± 0.21 ^bA^	8.48 ± 0.21 ^bA^
Bifidobacteria	0	7.34 ± 0.02 ^aA^	7.19 ± 0.09 ^aA^	7.30 ± 0.08 ^aA^
	6	9.19 ± 0.18 ^bA^	9.13 ± 0.12 ^bA^	9.19 ± 0.18 ^bA^
	24	9.26 ± 0.18 ^bA^	8.84 ± 0.51 ^bA^	9.43 ± 0.18 ^bA^
Clostridia	0	6.72 ± 0.87 ^aA^	6.35 ± 0.06 ^aA^	6.44 ± 0.31 ^aA^
	6	7.14 ± 0.79 ^aA^	6.19 ± 0.19 ^aA^	7.65 ± 0.38 ^bA^
	24	8.37 ± 0.11 ^aAB^	8.19 ± 0.08 ^bA^	8.68 ± 0.18 ^bB^
Enterobacteria	0	6.38 ± 0.07 ^aA^	6.29 ± 0.10 ^aA^	6.31 ± 0.15 ^aA^
	6	9.16 ± 0.18 ^bA^	9.16 ± 0.09 ^bA^	8.94 ± 0.53 ^aA^
	24	9.25 ± 0.16 ^bA^	8.63 ± 0.51 ^bA^	8.72 ± 1.18 ^aA^
Lactobacilli	0	6.46 ± 0.56 ^aA^	5.16 ± 0.20 ^aA^	6.41 ± 0.55 ^aA^
	6	7.44 ± 0.21 ^aA^	6.27 ± 0.30 ^aB^	7.40 ± 0.22 ^aA^
	24	7.28 ± 0.19 ^aA^	6.68 ± 0.74 ^aA^	7.40 ± 0,05 ^aA^
*L. reuteri* INIA P572	0	6.28 ± 0.62 ^a^	–	6.27 ± 0.54 ^a^
	6	7.10 ± 0.43 ^b^	–	7.17 ± 0.23 ^b^
	24	7.00 ± 0.52 ^ab^	–	7.07 ± 0.26 ^b^

Data represent mean ± SD (*n* = 6). Values within the same sub-column (faecal microbial group/*L. reuteri* INIA P572) with different lower-case superscript letters (^a, b^) indicate statistically significant differences (*p <* 0.05) for a given batch fermenter. Values within the same row with different upper-case superscript letters (^A, B^) indicate statistically significant differences (*p <* 0.05) for a given time. Values for *L. reuteri* INIA P572 within the same row did not show significant differences (*p* < 0.05) at any time.

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
