# Peer review of "Probiotic and Functional Properties of Limosilactobacillus reuteri INIA P572"

_nutrients, 2021, doi:10.3390/nu13061860_

Round 1
Reviewer 1 Report
This study is to evlatue the effect of Limosilactobacillus reuteri INIA P572 using in vivo and in vitro mouse model.
In addition, the authors looked at immunomodulatory effects of the strain during inflammation condition.
The conclusion was that antimicrobial components including reuterin that were produced by the probiotic strain could improve inflammatory condition in a DSS-induced mice model.
It is very well written MS and all results indicated in this study are clearly consistent with author's hypothesis.
It is a really interesting study having a in vivo and in vitro works.
I have enjoyed reading this MS.
However, minor points should addressed before it is considered as a publication.
Lin 49-50: explain briefly about iron absorption. how dose it work and associate with gut health?
Line 78-86, 454: whether this strain name (L. reuteri INIA P572) should be italic or not. Please check this through MS.
Line 79-81: any condition using shaking incubator before added on the plate.
Line 142: what do you mean the cell activation? adherent cell culture?
Line 181-182: How did you store the strain before administered to animal vehicle?
Line 218: check the section number.
Line 383-384: please specify why viable cells have more advantages based on what?
Figure 4: Is it the same resolution in three images? Please indicate the scale bar.
Line 482-483: As a future perspective point, you may suggest possible interaction between different bacteria species may have synergistic effects or not.
Author Response
REVIEWER #1
Comments and Suggestions for Authors
This study is to evaluate the effect of Limosilactobacillus reuteri INIA P572 using in vivo and in vitro mouse model. In addition, the authors looked at immunomodulatory effects of the strain during inflammation condition. The conclusion was that antimicrobial components including reuterin that were produced by the probiotic strain could improve inflammatory condition in a DSS-induced mice model. It is very well written MS and all results indicated in this study are clearly consistent with author's hypothesis. It is a really interesting study having a in vivo and in vitro works. I have enjoyed reading this MS.
We would like to thank to the reviewer for these comments.
However, minor points should addressed before it is considered as a publication.
Lin 49-50: explain briefly about iron absorption. how dose it work and associate with gut health?
A brief explanation about how reuterin inhibit iron absorption has been added to the revised manuscript according to the reviewer’s comment. The information about its potential therapeutic roles in diseases of iron overload are explained in the reference 22.
“Recently, it was also demonstrated that reuterin from L. reuteri is a functional gut metabolite able to modulate host iron absorption by suppressing intestinal hypoxia-inducible factor 2α activity [22]”
Line 78-86, 454: whether this strain name (L. reuteri INIA P572) should be italic or not. Please check this through MS.
- reuteri must be in italic, but not the strain name. In previous publications it has been always referred as that.
Line 79-81: any condition using shaking incubator before added on the plate.
Shaking incubation is not necessary at all to propagate L. reuteri.
Line 142: what do you mean the cell activation? adherent cell culture?
Cell activation means the activation of the RAW264.7 cell with L. reuteri / LPS. In order to avoid any confusion the word “activation” has been changed for “stimulation”. RAW 264.7 is an adherent cell line and must be scraped and counted before performing the assays in 96-well plates.
“Before cell stimulation, RAW 264.7 cells were scraped, cell suspension was adjusted to 106 cells/ml and 200 µl of the suspension was incubated…”
Line 181-182: How did you store the strain before administered to animal vehicle?
The strains were kept at -20ºC and thawed before the administration to mice. The cfu/ml of the milk suspension was checked after thawing.
Line 218: check the section number.
We are sorry for the mistake, section number “2.6.4. Flow cytometry” has been corrected in the revised manuscript.
Line 383-384: please specify why viable cells have more advantages based on what?
Although it is true that non-viable cells can have an impact in terms of mucosal immune stimulation, the beneficial effects of probiotics depend greatly on their viability during the passage through the gastrointestinal tract. In fact, probiotics are defined as live microorganisms that, when administered in adequate amounts, confer a health benefit on the host. To avoid any misunderstanding in the sentence it has been modified according to the reviewer’s comment:
“This is crucial to impact the balance of the intestinal microbiota [50] and, in many cases, also to confer stimulation of the intestinal immunity [51]
Figure 4: Is it the same resolution in three images? Please indicate the scale bar.
Yes, the three images had the same resolution. According to the suggestion made by the reviewer, the scale bar has been included in each figure
Line 482-483: As a future perspective point, you may suggest possible interaction between different bacteria species may have synergistic effects or not.
Following the suggestion made by the reviewer, a new sentence about with this suggestion has been added in the revised manuscript in the Conclusion section.
“This strain may be also suitable for developing novel multi-strain probiotic food products with complementary positive effects in both food safety and maintaining gut homeostasis”
Reviewer 2 Report
Its an interesting study and author has tried to address all aspect of the Limosilactobacillus reuteri INIA P572. Here are few suggestions :
- Please go through the manuscript for typos.
- Author has used lower case a and b, it is not clear what it indicates. I would suggest in legend you can mention b p<0.05, b = xxx vs ssss. Ensure to be more clear, even if you need to reiterate do mention it clearly in all legends. In certain figures with no effect the lower case, so it's confusing. Only indicate when there is an effect not otherwise, as it's distracting and misleading.
- Author should not leave control in figures, mention the control group for instance in figure 2.
- Figure 1, A & B represent the same data showing how in one the effect between 1ug/ml LPS vs 1ug/ml LPS + L.reutri is so significant (A), the same effect is gone in % measure (B). Also, be consistent the way you present data.
- Author mention microscopic score, pls do mention units x per high power field (hpf).Also it’s a normal practise to have three image per section and images inserted lack scale units, normally images in DSS are of magnification of 200x , these images are at 100x. It's not even clear to see the mucus layer and other sections in such zoomed out images and do label d images if you want to show the change. Please be consistent throughout the image section which you like to show as it's easy to compare.
- Please do repeated anova values in the result section for data shown in figure 3. Its would give best assessment of data over time.
- In the quantification data for rt PCR, GAPDH was used as house keeping gene. Study has shown GAPD is known to be less stable gene especially when considering the effect of immune markers like TNF and so on mouse model of DSS (https://www.ncbi.nlm.nih.gov/pmc/articles/PMC5301225/)
Author Response
REVIEWER #2
Comments and Suggestions for Authors
It is an interesting study and author has tried to address all aspect of the Limosilactobacillus reuteri INIA P572.
We would like to thank to the reviewer for the comment.
Here are few suggestions:
- Please go through the manuscript for typos.
We are sorry for the typographical errors. The manuscript has been revised and the typos corrected
- Author has used lower case a and b, it is not clear what it indicates. I would suggest in legend you can mention b p<0.05, b = xxx vs ssss. Ensure to be more clear, even if you need to reiterate do mention it clearly in all legends. In certain figures with no effect the lower case, so it's confusing. Only indicate when there is an effect not otherwise, as it's distracting and misleading.
Legend of Table 2 has been changed accordingly. Although we understand that the number of letters could be misleading, we think that we should add all the statistical data in the table and not only the significant differences.
“Data represents mean ± SD (n=6). Values within the same sub-column (faecal microbial group / L. reuteri INIA P572) with different lower-case superscript letter indicate statistically significant differences (P < 0.05) for a given batch fermenter. Values within the same row with different upper-case superscript letter indicate statistically significant differences (P < 0.05) for a given time.”
- Author should not leave control in figures, mention the control group for instance in figure 2.
Following the suggestion made by the reviewer, the term “control” has been added to figure 2 to improve the clarity.
- Figure 1, A & B represent the same data showing how in one the effect between 1ug/ml LPS vs 1ug/ml LPS + L.reutri is so significant (A), the same effect is gone in % measure (B). Also, be consistent the way you present data.
-We are sorry for the confusion, but there was a mistake in Fig. 1A. Thus, “LPS 1µg/ml” under “L. reuteri“ has been removed.
- Figure 1B represents the potential effect of the pre-stimulation of RAW 264.7 cells with L. reuteri INIA P572 against the challenge with 1µg/ml LPS. Although there was not significant difference in the NO production in comparison with those cells treated only with LPS (it is true that these data are showed in Fig. 1A). This Figure shows (in % of NO) that pre-treatment with the probiotic does not have an in vitro effect on this parameter.
Some sentences have been changed to clarify this aspect:
L155: “NO production was studied in supernatants from stimulated RAW 264.7 cells with L. reuteri INIA P572 or with lipopolysaccharides (LPS) from Escherichia coli (Sigma) (0.01, 0.1 or 1 µg/ml) “
L292: “Immunomodulatory activity of L. reuteri INIA P572 was studied in RAW 264.7 cells by measuring NO production upon 24 h stimulation with the probiotic compared with different concentrations of LPS (Figure 1A).”
L299: “In addition, the effects of L. reuteri INIA P572 were also studied in RAW 264.7 cells stimulated with 1µg/ml LPS. The incubation with L. reuteri INIA P572 before LPS stimulation did not significantly modify NO production in comparison with those cells treated with LPS only (Figure 1B).”
L311: “However, the incubation of these cells with LPS resulted in significant increased levels of these cytokines in comparison with basal conditions (P < 0.05), which were increased when the cells were pretreated with L. reuteri INIA, although statistical differences were only obtained when IL-1β production was considered.”
5.Author mention microscopic score, pls do mention units x per high power field (hpf). Also it’s a normal practise to have three image per section and images inserted lack scale units, normally images in DSS are of magnification of 200x , these images are at 100x. It's not even clear to see the mucus layer and other sections in such zoomed out images and do label d images if you want to show the change. Please be consistent throughout the image section which you like to show as it's easy to compare.
According to the suggestion made by the reviewer, the figure 4 has been modified with images at higher magnification, and the scale included.
6.Please do repeated ANOVA values in the result section for data shown in figure 3. Its would give best assessment of data over time.
Following the suggestion made by the reviewer, the statistical analysis has been performed over time and the figure modified accordingly
7.In the quantification data for rt PCR, GAPDH was used as house keeping gene. Study has shown GAPD is known to be less stable gene especially when considering the effect of immune markers like TNF and so on mouse model of DSS (https://www.ncbi.nlm.nih.gov/pmc/articles/PMC5301225/)
We highly appreciate the reviewer’s comment; in fact, we are aware that housekeeping genes generally used for RT-qPCR data normalization could have some variations according to the tissue and model used. However, we did perform an initial evaluation when we established the DSS model in our lab, by measuring some of the most widely used housekeeping genes (including Gapdh, β-actin and Tbp). Since we found no differences between them (and their combination did not alter the results), and over the progression of inflammation up to 10 days after initiation of DSS administration, we decided to continue our analysis with Gapdh, as it is the most widely used housekeeping gene.